# Continuous Authentication against Collusion Attacks [note 1]

**DOI:** 10.3390/s22134711

**Published:** 2022-06-22

**Authors:** Pin Lyu, Wandong Cai, Yao Wang

**Affiliations:** 1School of Computer Science, Northwestern Polytechnical University, Xi’an 710129, China; caiwd@nwpu.edu.cn; 2School of Cyber Engineering, Xidian University, Xi’an 710129, China; wangyao@xidian.edu.cn

**Keywords:** continuous authentication, smartphone authentication, gait authentication, accelerometer, imitation attack

## Abstract

As mobile devices become more and more popular, users gain many conveniences. It has also made smartphone makers install new software and prebuilt hardware on their products, including many kinds of sensors. With improved storage and computing power, users also become accustomed to storing and interacting with personally sensitive information. Due to convenience and efficiency, mobile devices use gait authentication widely. In recent years, protecting the information security of mobile devices has become increasingly important. It has become a hot research area because smartphones are vulnerable to theft or unauthorized access. This paper proposes a novel attack model called a collusion attack. Firstly, we study the imitation attack in the general state and its results and propose and verify the feasibility of our attack. We propose a collusion attack model and train participants with quantified action specifications. The results demonstrate that our attack increases the attacker’s false match rate only using an acceleration sensor in some systems sensor. Furthermore, we propose a multi-cycle defense model based on acceleration direction changes to improve the robustness of smartphone-based gait authentication methods against such attacks. Experimental results show that our defense model can significantly reduce the attacker’s success rate.

## 1. Introduction

The principle of biometric authentication is to use an individual’s unique biological characteristics to determine whether someone is who they says they are. Biological characteristics include the human body’s inherent physiological characteristics (e.g., fingerprints, faces, and irises) and behavioral features (e.g., handwriting, voice, and gait) [1,2]. With the development of IoT devices, users are using wearable sensors for biometric authentication, which one of the most prevalent authentication methods. In the past ten years, mobile devices such as smartphones, tablet computers, and smartwatches have embedded more and more sensors. The smartphone-based authentication is also called "transparent, implicit, active, nonintrusive, nonobservable, adaptive, unobtrusive, and progressive" techniques [3,4,5].

The definition of gait authentication is a method of verifying an individual by the manner of walking. From a technological perspective, Gafurov [6] divided gait authentication into three categories, machine vision (MV) based [7,8], floor sensor (FS) based [9,10], and wearable sensor (WS) based [10,11,12]. The general approach to gait authentication consists of four steps: Placing the device in some place that can record walking motion and collect data; preprocessing the data to reduce noise during data collection and environmental factors; using gait cycles or machine learning algorithms for detection; an analysis process [5].

Although the gait authentication system provides much convenience, it also incentivizes attackers to explore its vulnerability and launch an attack. In addition to the schemes to prevent the system from running, the schemes based on disrupting the system’s operation mainly include: destroying the feature extraction program, the result calculation program, and tampering and injecting data into the communication and storage procedures between various components. In contrast, attacks that use fake biometrics to target sensors do not require much knowledge of the target system and are less complex to implement. In gait authentication systems, attackers often create fake biometrics by mimicry. Unlike other biometric features, the various data related to gait can be collected in public, for example, by stalking or video. Through video recording, following, walking together, and other means, malicious attackers can obtain the details of a person’s gait, such as the height of the foot from the ground, the state of foot fall, bending of the upper body, the orientation of elbow movement, and the speed and rhythm of walking. When attackers have a detailed gait data set, they may use such training to achieve an attack. In recent years, approaches based on gait authentication [13,14,15,16,17,18] have been increasing, so it is essential to ensure the robustness [13,19,20,21,22] of the authentication systems.

Depending on whether the attacker intends to spoof the systems, Gafurov [10,23] divides the attacks into two types: friendly and hostile. The friendly scenario, known as a zero-effort or passive attack [24], means the attacker and victim have similar biometric features when facing the authentication systems. The hostile scenario is also known as an adversary attack or active attack [24]; in this case, the attacker is trying to mimic the victim’s biometrics. By combining the two cases, we propose a novel imitation attack model, the collusion attack. Our attack model no longer uses the pattern of the attacker to mimic the victim’s gait but instead uses the exact gait specification to train all the attackers. We invited 20 participants from the honor guards; these people have similar physical conditions. We used the quantified gait instructions to train all the participants and conducted training over three months. This work complements the part about the failure to complete the zero-effort and minimum-effort attacks in mimic attacks [25] and our previous work [12]. Further, we used three state-of-the-art gait recognition schemes (dynamic time warping (DTW), support vector machine (SVM), and random forest (RF)) as a target system to test the performance of our attack.

After analyzing our attack model’s results, we study the reasons behind under-performance. We found that our attack can reduce the gap in acceleration values to a certain extent by analyzing the results. Our training instructions are quantified, with detailed requirements such as speed, stride length, and foot height. The participants build muscle memory for these movements without losing stability and improvisation problems in the previous work. Hence, the false match rate is improved. Then, we propose a new defense model, which does not introduce new sensors, and focuses on the acceleration direction change. The experimental results show that our method performs similarly to the multi-device multi-sensor solutions [22]. Furthermore, it is stable in multiple scenes.

This work primarily extends the idea presented in Lyu et al. [12]. The notable differences between this and the previous work are as follows: (1) The attack proposed by Lyu et al. [12] only tested the gait authentication system that uses DTW. We tested the collusive attack using three state-of-the-art authentication systems. The result shows that our attack can affect the performance of the systems that use acceleration values as a feature. In addition, this paper provides more details about the attack. (2) The defense model proposed by Lyu et al. [12] requires researchers extract the features manually and use the binary search algorithm to find a suitable pure quaternion to rotate raw data. This process is very inefficient for finding suitable three-dimensional data. Moreover, users need to place their smartphones facing a specific direction; therefore, we have adopted a new method, transforming the device’s coordinate system to the world coordinate system and finally obtaining the body coordinate system for conversion. This method is fast; although the position of the mobile phone will affect the performance to some extent, but not seriously. (3) This paper proposes a new multi-cycle-based approach instead of using only one cycle for authentication.

In general, the contribution of our article is as follows:We propose an effective attack model and verify that it can increase the false match rate of the target gait authentication system after training the participants. This work complements the part about the failure to complete the zero-effort and minimum-effort attacks in mimic attacks and our previous work.We propose a defense model that uses the direction change of acceleration as the main feature to improve the system’s robustness when facing our attacks.We implement a continuous authentication system and conduct experiments with various scenarios. The results show that our defense model performs similarly to the multi-sensor system.

The remainder of this paper is organized as follows. Section 2 surveys the relevant work, including attack models and sensor-based gait authentication and impersonation attacks. In Section 3, we introduce the preliminary background and motivation for our work. A collusive attack is shown in Section 4. Our defense approach is discussed in Section 5. Section 6 describes the experiments and results of our defense model. We discuss and conclude our work in Section 7.

## 2. Related Work

Human gait refers to a manner of walking, stepping, or running. Kinetic studies and clinical studies on gait systems began in the 1950s. Gait is a universal uniqueness [26], and according to that fact, we can extract gait features during walking. After classification and recognition, we can finally achieve the purpose of authentication or recognition.

### 2.1. Wearable Sensor-Based Gait Authentication

Gait authentication refers to verifying an individual by the style of walking. From a technological perspective, Gafurov [6] divided gait authentication into three categories: MV-based, FS-based, and WS-based. In 2005, Ailisto et al. [27] published their research on using a wearable sensor-based approach for gait analysis. It is the first work in this area to our knowledge. After that, researchers used many kinds of motion sensors (accelerometers [25], gyroscopic sensors [28], force sensitive resistors [29], and bend sensors [30] ) to collect the motion of specific body parts. Because different human limb movements are unique and have universality [6], researchers acquired data from different positions, such as inside a pouch [31], pants pocket [25], or in the hand [32].

Nowadays, smartphones have a variety of embedded sensors. The common sensors include motion sensors (such as gravity, accelerometer, gyroscope, and magnetometer), environmental sensors (such as light, temperature, barometer, and proximity), and position sensors (such as GPS and compass). Numerous studies have leveraged these sensors for user authentication. Since 2009, smartphone-based gait authentication has become a hot research area, and many researchers have made significant contributions [25,33,34,35,36,37]. Muaaz et al. [25,38] used a DTW and SVM to classify legitimate users using gait-based features. With the popularization of devices such as smartwatches and sports bracelets in recent years, authentication schemes that combine multiple devices have gradually emerged [13,22]. The ZEMFA [22] system uses smartphones and Android smartwatches to extract walking biometrics and authorize an authentication session. ZEMFA uses a random forest model and works with 336 features derived from eight sensors of each device to defend against the treadmill attack. Furthermore, in Qin’s work [39], five accelerometers fixed at five body positions were used. They propose a multi-sensor fusion network; the EER of the correct rejection of unknown users was up to 6%.

### 2.2. Imitation Attacks

In the last dozen years, most research on gait authentication using wearable sensors has focused on zero-effort attack scenarios. Only a handful of studies have focused on carrying out attacks through imitation.

Stang [40] recruited 13 students to participate in the imitation experiment. Each person tried to imitate the target template 15 times for 30 minutes. The attacker cannot see how the target is walking during the imitation process and can only see the real-time value of the current acceleration on the screen. The experimental results show that some imitators exceed the judgment threshold of 0.5, which means that the attacker can imitate the gait of the victim. Mjaaland [41] pointed out the disadvantages of the experiment: on the one hand, the experiment only collected five gait templates from one victim; on the other hand, the data sampling rate was too low, and it was not easy to form a stable gait during the five-second acquisition process.

Depending on whether the attacker intends to spoof the systems, Gafurov [10,23] divides the attacks into two types: friendly and hostile. In the friendly scenario, the attacker and victim have similar biometric features when facing the authentication systems. In the hostile scenario, the attacker tries to mimic the victim’s biometrics. The results show that the minimal effort imitation attack did not significantly increase the attacker’s chances of being accepted. They concluded that imitators with similar physical characteristics and the same gender might better be accepted.

Based on the predecessors’ work, Mjalaand et al. [41] divide the hostile scenes into short-term and long-term scenes according to the length of the imitation time. In the friendly scenario, they collected the regular walking patterns of 50 participants as a performance baseline. They selected one victim and six attackers in the short-term hostile scenario and made five imitation attempts. They chose one available participant for six weeks in the long-term hostile scenario and made 60 attempts. The experiment results showed that the six imitators could not breach their respective physiological boundaries. The attacker could learn some of the characteristics but ultimately failed to match all the traits. Some attackers’ imitation performance had instead worsened over time, with attackers reporting that imitation training made their walking unnatural. The results of the long-term hostile scenario show that there may be multiple plateaus in imitation but with uncertainties. The various plateau periods produced by the same attacker indicate that the imitator can match the target gait. Still, the authors cannot draw definitive conclusions due to high uncertainty and insufficient data.

In Muaaz’s study [25], they recruited five mime artists trained in mimicking body motions and body language. In contrast, participants in the victim group are normal smartphone users. They used two phases, the reenact phase and the coincide phase. During the reenact phase, the attacker follows behind the victim to observe and learn. During the coincide phase, the attacker and victim walk side by side. The authors concluded that it is difficult for people to imitate the gait of others. Further, in 29 percent of the attempts, the attackers lost regularity in their steps while imitating the victims.

The above experiments imply two underlying hypotheses. First, visually similar walks from different individuals will produce similar sensor readings. Second, attackers can learn and repeat the gait pattern of the victims through imitation. In contrast, Rajesh Kumar et al. [42] extracted the victim’s gait and then used a treadmill to adjust the attacker’s gait characteristics. With the assistance of devices, attackers might be able to learn, adapt, and repeat the targeted gait patterns more easily.

## 3. Preliminary and Motivation

This section first introduces two widespread attack approaches that do not disrupt the system’s regular operation and give our choices. Then we explain our motivations at a theoretical level. Finally, we use an example to prove the feasibility of our theory.

### 3.1. Attack Model

Our research focuses on the authorization system, so we ignore the attack scheme of bypassing the system. We divide the common attack models into two types:Data injection: The attacker can directly manipulate motion sensors on unrooted Android devices [43]. The attacker can feed false data to the target system via the remote control. Alternatively, the attacker can inject manufactured false data into the authentication system, or build a robotic arm [44] to generate motion signals outside the devices.Imitation:Imitation based on human: The attacker train imitators to imitate the gait pattern of a selected victim [23,40,41]. Some researchers use feedback-based mechanisms to improve the efficiency of training [25]. Furthermore, some people use a machine, for example, a treadmill [22,42].

In summary, these attacks aim to obtain a similar input for the authentication system. Our research is more concerned with imitation attack scenarios.

### 3.2. Motivation

Mjaaland [45] proposes “Sheep” and “Wolf” characteristics in gait biometrics. “Sheep” refers to a person whose gait is easier to imitate in an imitation attack, and “Wolf” refers to a person who is better at impersonating the gait of others. That work inspires us: to find a “Sheep” person or find a “Wolf” person, or find both of them. Muaaz [25] attempted the “Wolf” side and used professional action actors; however, the attack effect was not noticeable due to the problem of improvisation on walking.

Another inspiration for us comes from limit theory in mathematics. An example is as follows:(1)$$\left.\begin{array}{c} \lim_{n\to \infty }X_{n}=T\\ \lim_{n\to \infty }Y_{n}=T\\ \forall \varepsilon >0\\ \exists N \end{array}\right\}\Rightarrow if\;n>N,\;\lim_{n\to \infty }\left|X_{n}-Y_{n}\right|<\varepsilon$$

In Equation (Equation 1), $X_{n}$ and $Y_{n}$ represent the gait sequence of two attackers while learning the target gait *T*. The larger subscript *n* indicates a later data collection time. In the imitation attack scenario, the Equation (Equation 1) represents that if attackers can imitate the “Sheep”, then the “Fake-sheep”’s gaits are similar.

Based on the above theory, we may use two people or more with similar gaits as the attackers and the victims to evaluate the target system.

### 3.3. Feasibility Analysis

We first implemented a gait authentication model based on Muaaz’s approach [25]. Then, we used the OU-ISIR dataset [46] to evaluate the system’s performance as the baseline. Finally, the EER that we obtain is 14.9%.

Next, our goal was to find people who can walk in the same style. There are two choices for us: elementary school students and military soldiers. Considering the stability of the gait and the improvisation issues mentioned earlier, we finally chose to recruit participants from the honor guard (detailed information about the participants are shown in the next section).

Honor guards often need to walk in uniform gait and march in formation. We use the authentication system to test their goose step; at last, the EER we obtain is up to 46%. It can satisfy our hypothesis that we can use the same gait from different people to evaluate an authentication system.

## 4. Collusive Attack

In previous studies, researchers have often used the attack model in which attackers imitate how the victim walks. The purpose of an imitation attack is that an attacker successfully mimics the victim’s gait. Its essence is to make the target system unable to decide whether the gait data are from the legal user. Hence, we propose a collusive attack model: the victim colludes with the attacker to provide a gait template that is easy to imitate; then, the attacker will easily use the same gait to obtain the wrong authorization.

### 4.1. Gait Training

As mentioned in Section 3.3, we invited 20 participants from the honor guards. In the honor guard, they went through significant goose-step training. They split the goose-step into individual movements during the training and stop at the end of each movement. After long-term training, they formed muscle memory of each movement, such as the distance between the hand and the body, the height of the foot from the ground, marching speed, and the arm swing speed. In addition, since honor guards tend to recruit members who meet the specified criteria, our participants have the same gender with similar ages, heights, and weights.

We designed our walking style according to the walking manner of the goose-step and the ordinary gait. We made the participants train individually for one hour a day for three months, with joint training each Monday. In addition, we asked participants to follow other participants in their lives when they met others (see Figure 1).

### 4.2. Performance of Collusive Attack

Figure 2 shows the distance distributions between the attackers and victims in all attempts. The horizontal axis represents the distance between the participant’s gait and the template. The vertical axis represents the distribution density. Since we had a 19:1 ratio of attackers to victims in each attack attempt, we replaced data frequencies with data distributions. From Figure 2, we can see a distinct overlap between the attacker’s performance and the victim’s.

Figure 3 shows the DET curve for three datasets. The blue line is the performance baseline when using Muaaz’s model (mentioned in Section 3.3) on the OU-ISIR dataset (EER=14.9%). The green line is the performance on the goose-step dataset, and the EER exceeds 45%. After analysis, we believe a possible reason that the goose-step-style usually requires honor guards to move quickly and violently to ensure it is worth watching. From the dataset, the acceleration value changes remarkably quickly and pauses at the end of the action. The DTW scheme using distance measurement does not reflect the differences between different individuals. Our attack (the orange line) achieved an EER of 39%, an unacceptable value. That is to say, the attack methods proposed in our work are more effective than mimic attacks. The under-performance of the baseline method verifies the effectiveness of our collusion attack.

In addition, we tested another two widely used classifiers: SVM and random forest (RF). We chose the Radial basis function (RBF) as the kernel for the SVM model [47] and used 100 as the cost value and γ = 0.01. We used the five features of the dominating feature set from Kumar’s work [42] for the RF model. The results show that the recall and precision of these two classifiers are less than 80%. According to the results, we believe that the following aspects may have played a vital role in improving the mimic effect:

Muscle memory. After a long-term training, each participant built a muscle memory of the training gait; therefore, participants will no longer have problems with irregular movements and improvisation during walking.

Detailed instructions.The gait details used in training are all quantified. In addition to regular stride and speed, we also quantify the upper body’s movements. Compared with imitating a new gait by visual observation, our approach enables participants to learn a new gait in a more stable process.

Multiple training types.The training process includes single and collective training, avoiding mutual compromise during joint training [48,49].

## 5. Our Defense Model

In this section, we introduce our countermeasure and its components and algorithms.

### 5.1. Approach Overview

The primary goal of our defense model is to resist collusion attacks, and the secondary goal is to use fewer resources. After several attempts, we finally chose a scheme that only uses an accelerometer.

Similar to the general approach to gait authentication, our defense model consists of four steps: Placing the device in the front right pocket of the trousers to record walking motion and collect data; preprocessing the data to reduce noise during data collection and environmental factors, and divide the data into cycles; using gait cycles compare with the templates; an analysis process.

### 5.2. Data Preprocessing

The data preprocessing module takes place to clean, reduce the noise, and normalize the data into cycles.

Walking data extraction.After obtaining the raw data, the first step is to extract the motion data. We use the sliding window to remove data from non-motion phases. Based on our previous experience, our study used 250 sample points as the window width and 16 as the threshold for acceleration. When the values of the five consecutive windows exceed the threshold, we consider this to be the walking phase. Conversely, when the five consecutive window thresholds are below the threshold, we consider this is a non-walking phase.

Noise removal.Since the gait-related features depend on the readings from the accelerometer, and these sensors are susceptible, we must consider the effects of noise when using the data. Typically this type of noise can be handled using linear interpolation and filtering techniques. This paper used Savitzky–Golay [50] smoothing filter (SG filter) to remove noise from the data. Compared to other moving average filters, SG filters reduce noise and maintain the shape and height of the waveform peaks. We calculate the smoothed value of a sample point as follows:(2)$$Y^{\prime }=X\cdot {\left(X^{T}\cdot X\right)}^{-1}\cdot X^{T}\cdot Y$$

In Formula (Equation 2), $X\cdot {\left(X^{T}\cdot X\right)}^{-1}\cdot X^{T}$ is the convolution coefficient, *Y* is the observation value, and $Y^{\prime }$ is the smoothing result.

Cycle extraction.Two stages conduct the cycle extraction process. First, we need to calculate the cycle length and then extract the cycle data based on it.We use 200 consecutive sample points as a window, move the window forward by one sample, and calculate the Euclidean distance between each corresponding point and sum. By repeating this process, we can obtain a sequence of distances. The number of sampling points between the local minima is the length of the cycles.

Once we have the cycle length, we start extracting the cycle data. Since the number of sample points in each cycle is mostly different, we add an offset value to the previous period length and use a period length of 1.2 times to find the local minimum. This local minimum is used as the starting point of the period to extract the period data.

Gravity separation.The direction of gravity relative to the smartphone’s coordinate constantly changes during walking. Since the value is too significant (approximately $9.8m/s^{2}$) to ignore, we need to eliminate the contribution of gravity. Android applies a high-pass filter (See Equation (Equation 3)) to achieve that and a low-pass filter (see Equation (Equation 4)) to isolate the force of gravity. Among them, $A_{x}\left(t_{n}\right)$ is the value of the accelerometer at time $t_{n}$ on the x-axis, $G_{x}\left(t_{n}\right)$ is the value of gravity at time $t_{n}$ on the x-axis, and three $L(t_{n})$ is the linear acceleration we need α=t/(t+dT) with *t* is the low-pass filter’s time-constant, which is the single sampling time of the sensor, and dT is the sampling frequency.
(3)$$\left\{\begin{array}{cc} \; & G_{x}\left(t_{n}\right)=\alpha G_{x}\left(t_{n-1}\right)+\left(1-\alpha \right)A_{x}\left(t_{n}\right)\\ \; & G_{y}\left(t_{n}\right)=\alpha G_{y}\left(t_{n-1}\right)+\left(1-\alpha \right)A_{y}\left(t_{n}\right)\\ \; & G_{z}\left(t_{n}\right)=\alpha G_{z}\left(t_{n-1}\right)+\left(1-\alpha \right)A_{z}\left(t_{n}\right) \end{array}\right.$$
(4)$$\left\{\begin{array}{cc} \; & L_{x}\left(t_{n}\right)=A_{x}\left(t_{n}\right)-G_{x}\left(t_{n}\right)\\ \; & L_{y}\left(t_{n}\right)=A_{y}\left(t_{n}\right)-G_{y}\left(t_{n}\right)\\ \; & L_{z}\left(t_{n}\right)=A_{z}\left(t_{n}\right)-G_{z}\left(t_{n}\right) \end{array}\right.$$

Abnormal cycles removal.After obtaining the gait cycles, we also need to remove the abnormal cycles. Because some unexpected situations may occur, e.g., stumbled during the data acquisition process. We used DTW [25] to calculate distances between paired cycles. After summing, we removed the cycle that was 50% more than the minimum distance.

### 5.3. Coordinate Transformation

During walking scenarios, the data are recorded according to the device coordinate system. Instead of aligning the coordinate system of data with the template’s coordinate system [12], we use the device’s body inclination angles to transform the data. There are advantages of this method, including:1.The device position in pocket can affect the data to some extent, but not seriously;2.The diversity in user walking and device motion is the main feature of our study, the transform cannot affect the evaluation;3.The different users have similar device-to-body-angle ratios.

We use quaternions to calculate coordinate rotations. These can avoid the problem of gimbal lock and are simpler than Euler angles, and they are numerically stable and more efficient compared to rotation matrices. Quaternions are in the form of q=a+bi+cj+dk, where a, b, c, and d are real numbers; i, j, and k are the basic quaternions. We convert the raw data to the data in the new coordinate system using quaternion:(5)$$P_{w}=q_{\mathrm{sw}}P_{s}q_{\mathrm{sw}}^{-1}$$
where $P_{s}$ is the raw data collected from the device, $P_{w}$ is the rotated data in the world coordinate system. The quaternion $q_{\mathrm{sw}}$ is obtained directly from the device, and $q_{\mathrm{sw}}^{-1}$ is the conjugate quaternion of $q_{\mathrm{sw}}$.

When collecting walking data, participants would face different directions. Just using data in the world coordinate system cannot provide stable walking patterns to achieve accurate sensing. Hence, we need to transform the data to normalize the sensor readings, computed as:(6)$$P_{f}=q_{\mathrm{wf}}P_{w}q_{\mathrm{wf}}^{-1}$$
where $P_{f}$ is the final data in the walking coordinate system. $q_{\mathrm{wf}}$ is the rotation quaternion, and the $q_{\mathrm{wf}}^{-1}$ is the conjugate quaternion of $q_{\mathrm{wf}}$. In this transform process, we use the Euler angle to calculate the quaternions $q_{\mathrm{wf}}$, because it cannot be obtained directly. The angles of rotation around the X, Y, and Z axes are θ, ψ, and ϕ, which are defined in:(7)$$q_{\mathrm{wf}}=\left[\begin{array}{c} sin\frac{\phi }{2}cos\frac{\theta }{2}cos\frac{\psi }{2}-cos\frac{\phi }{2}sin\frac{\theta }{2}sin\frac{\psi }{2}\\ cos\frac{\phi }{2}sin\frac{\theta }{2}cos\frac{\psi }{2}+sin\frac{\phi }{2}cos\frac{\theta }{2}sin\frac{\psi }{2}\\ cos\frac{\phi }{2}cos\frac{\theta }{2}sin\frac{\psi }{2}-sin\frac{\phi }{2}sin\frac{\theta }{2}cos\frac{\psi }{2}\\ cos\frac{\phi }{2}cos\frac{\theta }{2}cos\frac{\psi }{2}+sin\frac{\phi }{2}sin\frac{\theta }{2}sin\frac{\psi }{2} \end{array}\right]$$

In our study, participants are walking on a flat road. Thus, we assume θ and ψ are zero. In this case, ϕ can be defined as the counterclockwise rotation angle around the north direction in the world coordinate system. We calculate the accumulated distance from the acceleration along the X-axis and Y-axis. Then, we can obtain the angle α caused by walking movements as:(8)$$\alpha =\left|arctan\left(\frac{Accumulated\;Distance\;in\;Y-axis}{Accumulated\;Distance\;in\;X-axis}\right)\right|$$

The range of α is between 0 and π/2 in Equation (Equation 8). We use quadrant method to convert α to angle ϕ:(9)$$\phi =\left\{\begin{array}{c} \begin{array}{cc} \frac{3\pi }{2}+\alpha ; & if\;Q=1,\\ \frac{\pi }{2}-\alpha ; & if\;Q=2,\\ \frac{\pi }{2}+\alpha ; & if\;Q=3,\\ \frac{3\pi }{2}-\alpha ; & if\;Q=4, \end{array} \end{array}\right.$$
where *Q* is the quadrant of leg movement, we can estimate it on the order of peaks and through on accelerations on X-axis and Y-axis. At last we obtain the acceleration on three axes:(10)$$\left\{\begin{array}{cc} \; & A_{x}=arctan\left(\frac{\sqrt{a_{y}^{2}+a_{z}^{2}}}{a_{x}}\right)\\ \; & A_{y}=arctan\left(\frac{\sqrt{a_{x}^{2}+a_{z}^{2}}}{a_{y}}\right)\\ \; & A_{z}=arctan\left(\frac{\sqrt{a_{x}^{2}+a_{y}^{2}}}{a_{z}}\right) \end{array}\right.$$

### 5.4. Similarity Comparison

We use the force direction in three axes as an essential feature for identity verification and use the distance between cycles to measure the similarity.


**Direction extraction.**


At a sample point at time *t*, based on the three-axis readings read from the accelerometer, we can obtain the acceleration value *A(t)* for that point: $A\left(t\right)=\sqrt{A_{x}^{2}\left(t\right)+A_{y}^{2}\left(t\right)+A_{z}^{2}\left(t\right)}$. After that, we can obtain the direction on the three axes:(11)$$d_{x}=\frac{a_{x}}{A(t)},\;d_{y}=\frac{a_{y}}{A(t)},\;d_{z}=\frac{a_{z}}{A(t)}$$

Using Equation (Equation 11), we can obtain a sequence of vectors. In this sequence, the length of each vector is 1, and in Figure 4, we can see that these sample points are all distributed on a unit sphere. Each point on the sphere represents the direction of acceleration of that sample point, that is, the direction of the force at that time.


**Distance between cycles.**


Before calculating the distance during the week, we first introduce the calculation method of the distance between two sampling points. We use the arc length between two points on the sphere to represent the distance between two points. In a polar coordinate system, the arc length between two points is equal to the angle between two points. So we first use the following formula to calculate the cosine value of the angle:(12)$$\cos\left(\theta \right)=\frac{\vec{a}\cdot \vec{b}}{\left|\vec{a}\right|\left|\vec{b}\right|}$$

In Equation (Equation 12), $\vec{a}$ and $\vec{b}$ have lengths of 1, so we can obtain the distance between $\vec{a}$ and $\vec{b}$ by Equation (Equation 13):(13)$$dist\left(\vec{a},\vec{b}\right)=\theta =\arccos\left(\vec{a}\cdot \vec{b}\right)$$

In addition, according to our statistical results, the angle between two adjacent points is between 0 and 0.5π, because, based on our sampling rate, no one can swing their leg more than $90^{\circ }$ in such a short time.
(14)$$D\left(i,j\right)=dist\left(i,j\right)+min\left\{\begin{array}{c} D(i-1,j)\\ D(i,j-1)\\ D(i-1,j-1) \end{array}\right.$$

We use Equation (Equation 14) above to calculate the distance between two periods. A shorter distance means that the two cycles are more similar, and during the certification phase, if the distance falls below the threshold we set, a "legitimate" judgment is made.

Finally, we use the KNN algorithm to determine which cycles to maintain. First, we determine the center cycle with the smallest sum of distances from other cycles, excluding cycles with long distances (the default value is 450). In the registration phase, the system submits five shortest cycles from the center as user templates and the longest distance in the top 90% as the threshold. In the authentication phase, the system uses the algorithm and the preserved template to calculate the distance and make a judgment.

We use confidence scores to assess whether to authenticate the current user. We calculate the confidence score based on whether the distance between the committed cycle and the template is below the threshold. Moreover, if the confidence score exceeds 50%, we determine that the current user is legitimate.

## 6. Evaluation and Discussion

Our experiment used two OPPO-R9s, two MI8s, and one MI8 SE as devices to collect gait data; we also used twenty participants (mentioned in Section 4). We installed our application on these devices and collected acceleration values with a sampling rate of 200Hz. The gait authentication application will record two types of logs: event log and data log. The event log contains all the events of the registration and verification phases with the results and timestamps. The data log contains log files of raw data and template data. We asked all participants to keep the device in the front right pocket of their pants. Moreover, we asked them to walk in a straight line; the duration was not less than one minute each time. Finally, we achieved an EER of 8.03%.

### 6.1. Performance of Our Approach

Since all participants used the same gait in our experiments, we could pick the best performing victim–attacker pairs to evaluate our defense model.

Figure 5 shows the best-performing attacker–victim pairs. The blue curve represents the confidence score obtained by the victim, and the red curve represents the attacker’s performance. We can see that with 50% as the threshold, no attacker can reach the threshold. That is, our defense model can resist collusion attacks. In 7% of the scenarios, the victim’s gait failed to pass the validation. We checked the timestamps and found that most of this phenomenon mainly occurs at the end of the walking phase. One possible reason is that the participants must be prepared to remove the device near the end of walking, thereby losing gait stability. This phenomenon occurs less in the beginning stages of walking. One possible reason is that, after training, the participants became habituated to this gait and are capable of performing stably without adjustment.

In addition, we reanalyzed the raw data sequence and found that a small number of cycles were successful in the best-performing attacker–victim pair. The peak of the confidence score for successful attacks reached 55.2%; however, these cycles were removed as abnormal during the pre-processing phase and failed to enter the authentication phase. Next, we introduce our optimization scheme to deal with abnormal cycles.

### 6.2. Multi-Cycle Model

We optimized our defense model to gain a robust model to fight against our attack. We try to use continuous cycles to upgrade the model for gait authentication instead of one cycle. Figure 6 depicts the EERs under the different number of cycles. When we use only one cycle, the EER is the same as the former study. Moreover, the number of continuous cycles corresponding to the smallest EER is what we need.

From Figure 6, we can see that EER is not monotonically increasing or decreasing and reaches the lowest value when number=3. To find the reason for this phenomenon, we analyzed the change of TPR (true positive rate) and TNR (true negative rate) under the different number of cycles (see Figure 7).

Figure 7 depicts the change of the TPR and the TNR rate under the different number of cycles. We should consider the trade-off between TPR and TNR in an authentication system. Higher TNR indicates that the systems reject more attacks, which means higher security; however, the corresponding TPR reduces simultaneously, making the system difficult to use. In Figure 7, we can see that the number of cycles significantly affects the TPR and TNR. The decrease in TPR and the increase in TNR come with the number increase. Finally, we choose number=3 and design the three-cycle-based model.

### 6.3. Performance under Different Gait

As mentioned in Section 4, we recruited 20 participants. In addition to the gaits we designed above, we collected the participants’ goose-step and walking data. We collected ten sets of data for each participant’s gait and finally divided them into about 1200 samples (for each gait). We use 10-fold cross-validation and then take the mean to study the performance of our defense model. Table 1 shows the result, and we can see that our defense model can achieve better performance in the state-of-the-art multi-sensor scenarios reported in previous work.

## 7. Conclusions

Using a smartphone for gait authentication is a convenient method. This work expands on our previous idea [12]. Combining active and zero-effort attacks, we propose a novel attack scheme, a collusive attack. We ask the attackers and the victims to do their best to walk in the same manner, rather than attackers imitating victims. The result proved that our attack could successfully attack specific gait authentication systems. We propose a new multi-cycle scheme to defend against this attack and upgrade our application based on that. We achieved an EER of 7.32% in real-time authentication in the defense scenario, better than our previous work’s performance. Although the data used in the attack scenario only contain a few topics, our work complements the previous work on active attacks. The results demonstrate that high-intensity training may increase the similarity between an attacker’s and the target’s gait. In future work, we will examine why the new features are functional. After that, we need to try more models and features. Because from previous work, we know that distance-based schemes calculate more than a trained model. In addition, we want to determine the boundaries between “mimicable” and “unmimicable” in terms of gait.

## Figures and Tables

**Figure 1 sensors-22-04711-f001:**
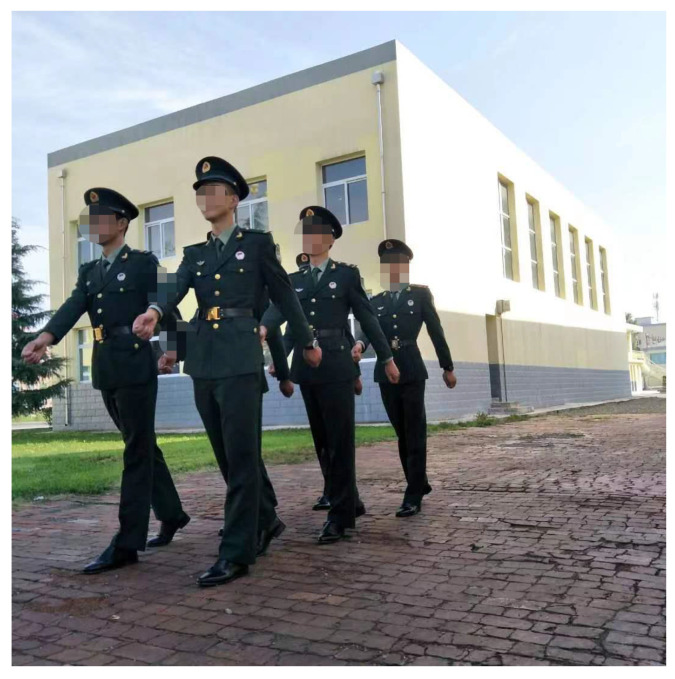
Walk together in a queue.

**Figure 2 sensors-22-04711-f002:**
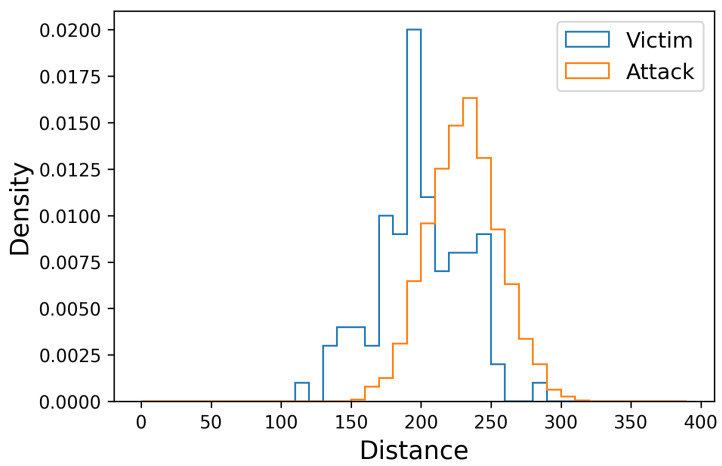
Distribution of the distance of attacker and victim from the victim’s gait template.

**Figure 3 sensors-22-04711-f003:**
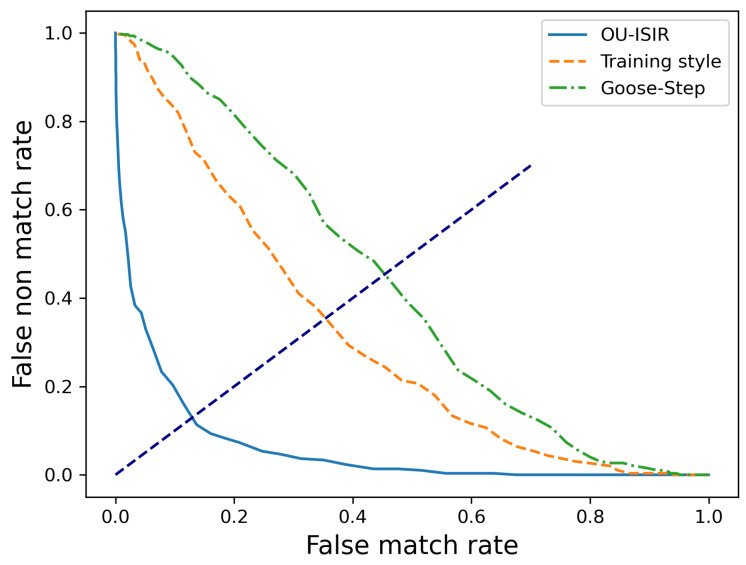
DET curve on different datasets; the threshold grow with 100 steps.

**Figure 4 sensors-22-04711-f004:**
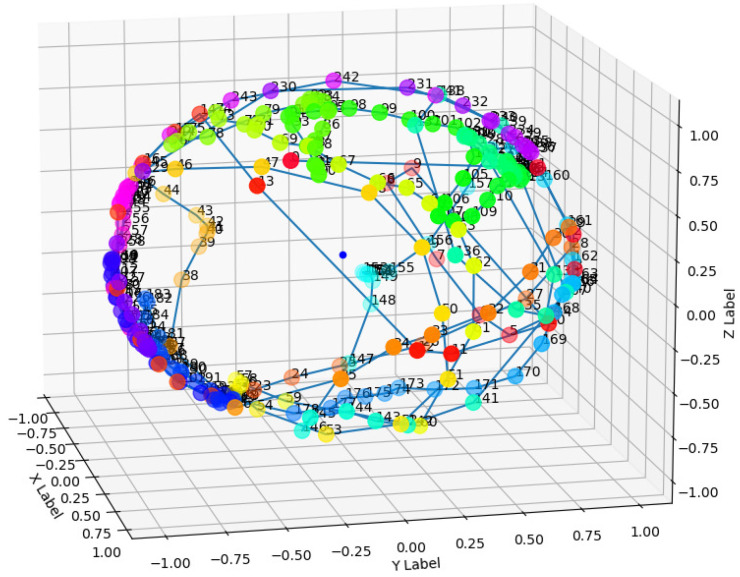
Direction of each sample points in a cycle.

**Figure 5 sensors-22-04711-f005:**
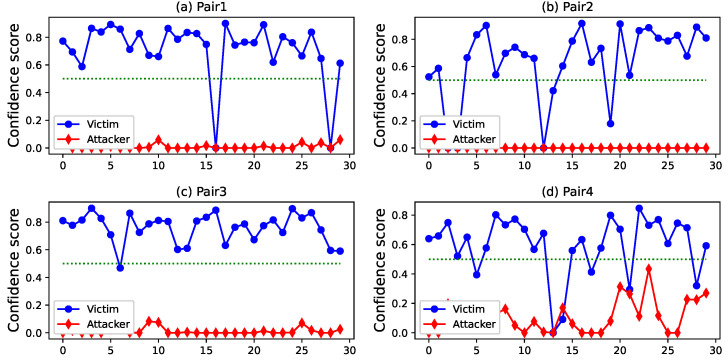
Best-performing attacker–victim pairs.

**Figure 6 sensors-22-04711-f006:**
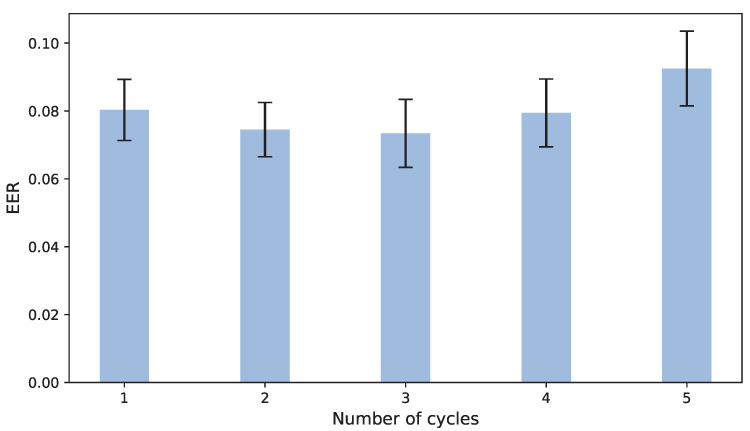
The average error rate under different number of cycles.

**Figure 7 sensors-22-04711-f007:**
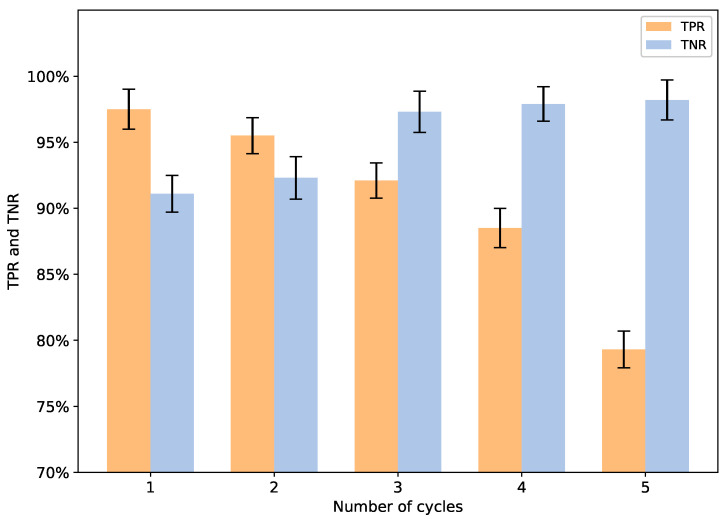
The true positive rate and the true negative rate under different number of cycles.

**Table 1 sensors-22-04711-t001:** Results of our approach.

Model Name	Training Style		Goose Step		Own Style	
	EER	Threshold	EER	Threshold	EER	Threshold
SVM	10.93%	0.48	12.79%	0.5	8.87%	0.74
Random Forest	13.91%	0.75	15.39%	0.41	9.98%	0.08
Our Model	7.32%	0.49	8.15%	0.51	7.95%	0.48

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
