# Peer review of "Continuous Authentication against Collusion Attacks†"

_sensors, 2022, doi:10.3390/s22134711_

Round 1

Reviewer 1 Report

This paper proposes a novel attack model to study the imitation attack in the general state and train participants with quantified action specifications. It is very interesting and could be published in Sensors. However, it needs to revise before accepted for publication. The comments are as follows:

1. Please describe the relation of your research about sensors topic in manuscript.

2. In the results of Fig. 3, the authors should explain in detail what are the meanings of the DET curve for three datasets.

3. Fig. 5 and Fig. 6 seem meaningless. The authors should explain the necessary of these two figures.

Reviewer 2 Report

This paper proposes a novel attack model called a collusion attack. However, I will comment on some aspects to improve the quality of the manuscript, and the changes made must be highlighted:

-Authors must avoid writing "he/her".

-The acorns are being used incorrectly. The correct way is to write its initial letter with capital letters of the meaning of the acronym, such as "Machine Vision (MV)". This error must be fixed in all acronyms throughout the document.

-The words Section, Figure, Algorithm, Table, Equation, are always written with the first letter in capital letters.

-In line 252, there is a missing space between the words.

-The axes of Figures 2 and 3 do not have the corresponding unit.

-The title of Figure 2 must be a title and not an explanation.

-Sections must have the corresponding verb tense.

-The authors do not present the data of the scenario and experimentation performed.

-Figure 9 must have the same size as the letters in the manuscript.

-The conclusions have to improve and add future work.

Author Response

Thank you for your work.

Reviewer 3 Report

The authors of this paper propose a novel attack model called a collusion attack. To this end, the authors study the imitation attack in the general state and its results and verify the feasibility of their attack. Secondly, they propose a collusion attack model and train participants with quantified action specifications. The results demonstrate that their attack indeed increases the attacker’s false match rate in some systems using acceleration.

The subject of the paper is quite interesting, well organized and presented. The results presented support the contribution of the paper. Overall, the reviewer believes that the paper can be accepted for publication. Some minor grammar/syntax errors can be corrected during the preparation of the camera ready version.

Author Response

Thank you for your work.

Best wishes.

Reviewer 4 Report

The work presents an attack model called collusion attack to attack gait authentication using a smartphone. Then, they present a defense model that can significantly reduce the attacker’s success rate. The work expands on a previous idea from the authors. In summary, a good paper that is easy to follow. Results look valid to me.

Author Response

Thank you for your work.

Best wishes.

Round 2

Reviewer 2 Report

Thanks to the authors for performing the changes suggested by the reviewers. Before the publication of the article, a minor spell check is required.

Author Response

Thank you for your work. We have fixed the problems.
